# Can co-ordination advancement of two-way FDI improve resource misallocation? Evidence from 285 cities in China

**Shuo Zhou[1], Suqing Shao** [ID] [2,3]*

1 School of International Trade and Economics, Anhui University of Finance and Economics, Bengbu, Anhui, China, 2 Business School, José Rizal University, Mandaluyong, Philippines, 3 Bengbu No.1 Middle School, Bengbu, Anhui, China

* shaosq5562@126.com

## Abstract

During the past 40 years of reform and opening-up, the implementation of an unbalanced enhancement strategy has led to severe resource misallocation, making the mitigation and rectification of this misallocation an urgent issue. This paper utilizes urban data from China between 2006 and 2020 to examine the impact of two-way foreign direct investment (FDI) on resource misallocation, as well as the pathways through which coordinated advancement of two-way FDI affects resource misallocation. After undergoing a series of robustness and endogeneity tests, the conclusion remains stable. Mechanism testing reveals that coordinated advancement of two-way FDI improves capital resource misallocation by enhancing the enhancement level of the financial industry, while exacerbating labor misallocation by increasing labor costs. This paper integrates the coordinated advancement of two-way FDI and resource misallocation into the same analytical framework, proposing policy recommendations to alleviate China's resource misallocation issue.

## 1 Introduction

Since the turn of the century, under the guidance of "bringing in" and "going out", the scope of China's outward foreign direct investment (OFDI) and international foreign direct investment in China (IFDI) has rapidly expanded. Based on the World Investment Report 2021, China utilized IFDI flows, reaching $144.4 billion, and OFDI flows, touching $153.7 billion, with IFDI and OFDI stocks accounting for 4.6% and 5.99% correspondingly by the end of 2020. Co-ordination advancement of two-way FDI is an important way of resource flow, connecting two markets and two types of resources, playing a crucial role in extending the industrial chain and promoting industrial upgrading. The co-ordination advancement of two-way FDI is conducive to the long-term sustainable enhancement of developing countries, and such inclusive growth will strongly promote the optimal allocation of resources and rapid economic growth [1]. China has guided IFDI into the market and led to a surge in labor demand and increased resource supply. At the same time, by integrating into the global industrial chain through OFDI, it guides domestic resources to flow abroad and affects the demand for domestic labor

---

**Data Availability Statement:** All relevant data can be found at the following location: DOI: 10.57760/sciencedb.09021 Description Can co-ordination advancement of two-way FDI Improve Resource Misallocation? Evidence from 285 Cities in China

Suqing Shao CSTR: 31253.11.sciencedb.09021
DOI: 10.57760/sciencedb.09021.

**Funding:** The authors received no specific funding for this work.

**Competing interests:** The authors have declared that no competing interests exist.

market. Therefore, it is inadequate to study the impact of IFDI or OFDI on resource misallocation solely from its perspective. It is crucial to combine both and study the co-ordination advancement of two-way FDI.

Due to the long-term imbalanced development model, the issue of resource misallocation in China has always been a focus of attention in the academic community. With the swift expansion of the economy, the coarse economic growth pattern has resulted in low efficiency in resource allocation [2]. The nature of the diffusion and transmission of resource allocation distortions has resulted in structural problems in Chinese economy [3], which deviating from the effective resource allocation and obstructing the China's economic advancement. The report of the 20th Party Congress states, 'Combining the implementation of the domestic demand expansion strategy with deepening of supply side structural reforms,' which is an unavoidable decision to hasten the formation of a novel enhancement model with a focus on the domestic macro-cycle and mutually reinforcing domestic and global double-cycles. In the context of double-cycling, the co-ordination advancement of two-way FDI promotes the optimal allocation of domestic resources [4]. Foreign and domestic multinational corporations invest their resources to where they yield the highest returns, forming the co-ordination advancement of two-way FDI. The co-ordination advancement of two-way FDI can integrate domestic and international resources, which can provide impetus for supply side structural reforms. In a narrow sense, resources are divided into capital and labor. The study of the misallocation of capital and labor resources due to the co-ordination advancement of two-way FDI is a question worthy of in-depth research.

Currently, scholars have dedicated significant attention to co-ordination advancement of two-way FDI and the problem of resource misallocation. Literature relevant to this paper encompasses two main areas: Firstly, the measurement of resource misallocation and the factors influencing them. Academics mainly measured resource misallocation in both micro and macro dimensions. At the micro level, there are mainly parametric and non-parametric methods. The parametric approach measure is to measure the extent of resource misallocation through the actual and effective output gap [5]. The non-parametric method combines resource flow with enterprise size and productivity level, and decomposes factor productivity through OP technology to obtain covariates that reflect the degree of resource misallocation [6]. At the macro level, Chen and Hu [7] used the absolute and relative deformation coefficients of capital prices to measure capital resource misallocation, providing scholars with ideas for later measuring resource misallocation; Bai and Liu [8] used urban data to construct the C-D production function and used regression method to calculate the capital resource misallocation and labor resource misallocation exponent at the China's urban level. Scholars have primarily measured capital resource misallocation and labor resource misallocation. The main drivers of capital resource misallocation are monopoly frictions and financial frictions [9], the enhancement of information technology [10], administrative friction [11], financing constraints [12], information asymmetry [13], stakeholder monitoring of firms [14]; The main driving factors driving labor misallocation are monopoly frictions [15], High-speed railway opening [16], labor skill levels [17], decision-making in the marketplace [18], increased Regulation of Businesses [19]. Secondly, the measurement of the degree of co-ordination advancement of two-way FDI and its implications. There are two existing measures in the literature to calculate the level of the co-ordination advancement of two-way FDI. Firstly, is to adopt the co-ordination advancement of two-way FDI interaction term, secondly is to calculate the degree of the co-ordination advancement of two-way FDI. The impact of the co-ordination advancement of two-way FDI, focusing mainly on green enhancement [20], green technology innovation efficiency [21], carbon emission reduction [22], quality economic growth [23]. From the above, extensive literature has been developed on resource misallocation and the co-

ordination advancement of two-way FDI. However, there is no literature that incorporates the co-ordination advancement of two-way FDI and resource misallocation into the same theoretical framework.

In light of this, the paper empirically examines the impact of the co-ordination advancement of two-way FDI on resource misallocation using panel data from 285 prefectural-level cities in China spanning from 2006 to 2020. Striving to contribute to enriching related research and offering reasonable policy recommendations. Moreover, the endogeneity issue was addressed by employing instrumental variable method and system GMM model. Furthermore, the study divided the stages of the co-ordination advancement of two-way FDI based on its level, exploring the impact of different stages of the co-ordination advancement of two-way FDI on resource allocation; And the impact of coordinated advancement of two-way FDI on resource misallocation was examined based on the enhancement levels and geographical locations of various cities. Finally, the transmission mechanism of the impact of the co-ordination advancement of two-way FDI on resource misallocation was discussed through the pathways of deepening financial enhancement and rising labor costs.

The results show that the co-ordination advancement of two-way FDI optimizes capital allocation by enhancing the level of financial enhancement. However, by increasing labor costs, it exacerbates labor misallocation. Despite a series of robustness checks, the results remain stable. Based on empirical evidence, the conclusion is drawn: China is currently in a high-quality enhancement stage transitioning from factor input to improving resource allocation efficiency. It should unwaveringly adhere to its strategy of opening-up to the outside world, encourage enterprises to actively 'bring in' and 'go out,' and strengthen macro-control. Relying on the co-ordination advancement of two-way FDI to achieve optimized resource allocation and high-quality economic enhancement.

Distinguished from previous research, the main contribution of this study may lie in: Firstly, integrating the co-ordination advancement of two-way FDI with resource misallocation into the same theoretical framework enriches the relevant research; Secondly, overcoming endogeneity issues through the use of instrumental variable methods and the system GMM model; Thirdly, conducting heterogeneous analysis from different perspectives to ensure the robustness of the results, which contributes to more precise formulation of policy pathways for improving resource misallocation; Fourthly, the utilization of a mediation effects model to examine the impact of the co-ordination advancement of two-way FDI on resource misallocation, leading to a deeper understanding of the underlying causes of resource misallocation.

The paper is structured as follows: the second part contains discussion and hypotheses, followed by analytical methods and data collection in the third part, the empirical testing steps, and analysis in the fourth part, the fifth part is mechanism analysis, and the finally, conclusions and recommendations in the last part.

## 2 Theoretical analysis and research hypotheses

Misallocation refers to a deviation from the ideal state of resource configuration. Unlike efficient resource allocation, 'resource misallocation' denotes the presence of barriers to the free allocation of different resource elements between regions and industries, hindering the achievement of optimal allocation. Cheng [24] classified resource misallocation into "internal mismatch" and "external mismatch". Bai and Liu [8] argue that varying degrees of capital resource misallocation and labor resource misallocation exist in different regions of China, with significant regional disparities. The following theoretical perspective analyses the effect of the co-ordination advancement of two-way FDI on resource misallocation.

In recent years, foreign direct investment (FDI) and outward foreign direct investment (OFDI) have experienced rapid growth as two major strategic pathways for China to acquire funds, technology, and market resources. The trend of co-ordination advancement of two-way FDI is increasingly prominent, serving as a significant driving force in guiding the rational flow of resources, improving resource misallocation, and enhancing resource allocation efficiency. Specifically, the co-ordination advancement of two-way FDI can improve capital, with the main pathway being the level of financial enhancement. On one hand, during the early stages of enhancement in developing countries, IFDI can alleviate the host country's capital needs and bring advanced technology and management expertise [25]. Through vertical and horizontal effects, it helps address financing constraints for enterprises in the host country, thereby reducing financing costs, enhancing capital liquidity, fostering a more open, inclusive, and efficient financial system, consequently improving capital resource misallocation, and enhancing capital allocation efficiency [26]. On the other hand, capital inflows also provide a solid foundation for outward investments [27]. The enhancement of the host country's financial strength and the improvement of its financial enhancement level create necessary conditions for enterprises to engage in cross-border capital allocation and operations. Through outward foreign direct investment (OFDI) reverse technology spillover effects, financial institutions in the host country have established close connections with those in developed countries. Through cooperation, exchange, and competition with financial institutions in these developed countries, the management level and operational efficiency of financial institutions in the host country have been significantly enhanced. Therefore, the interaction between foreign direct investment (FDI) and outward foreign direct investment (OFDI) continuously promotes and guides the improvement of the host country's financial enhancement level. The improvement of the financial enhancement level can increase the liquidity of resources, reduce transaction costs of financial investment, thereby increasing investment, effectively alleviating capital resource misallocation, and enhancing capital allocation efficiency [28].

**H1:** The co-ordination advancement of two-way FDI can enhance capital, primarily through advancements in the level of financial enhancement.

The co-ordination advancement of two-way FDI exacerbates the mismatch of labor resources by increasing labor costs. On one hand, IFDI increases the demand for labor in the host country, leading to changes in the geographical distribution of labor and exacerbating the issue of labor mismatch. At the same time, foreign enterprises will attract sustained labor inflows by raising labor prices [29]. This has led to a phenomenon of 'labor shortages' among local enterprises, exacerbating labor scarcity. With the continuous influx of IFDI, the host country may face challenges of labor supply shortages, further exacerbating resource misallocation. On the other hand, with the continuous outflow of OFDI, it may lead to the emigration of labor from some developing countries. This emigration of labor mainly manifests in two aspects. Firstly, some enterprises will export a portion of their labor force through outward foreign direct investment [30]. Secondly, outward foreign direct investment companies will dispatch management and technical personnel to overseas subsidiaries, inevitably creating some low-skilled job positions. This, in turn, further exacerbates the host country's demand for labor. Due to the inability of labor supply to meet rapidly growing demand in the short term, the rise in labor costs is actually detrimental to addressing the issue of labor mismatch. Therefore, two-way FDI exacerbates the degree of labor mismatch.

**H2:** The co-ordination advancement of two-way FDI exacerbates labor mismatch, primarily through increasing labor costs.

# 3 Research design

## 3.1 Model settings

Drawing on Bai and Liu (2018) [8], the formula is developed in the study to co-ordination advancement of two-way FDI's influence on capital resource misallocation and labor resource misallocation:

$$abstauk_{it} = \alpha + \beta dioc_{it} + \tau X_{it} + \delta_i + \sigma_t + \varepsilon_{it} \tag{1}$$

$$abstaul_{it} = \alpha + \beta dioc_{it} + \tau X_{it} + \delta_i + \sigma_t + \varepsilon_{it} \tag{2}$$

Here, i denotes the city and t denotes the year; abstauk_{it}, abstaul_{it} are the explained variables, denoting the capital resource misallocation and labor misallocation indices for city i in year t, respectively. The core explanatory variable is expressed as the coupled coordination exponent of two-way FDI. $X_{it}$ denotes control variables, $\delta_i$ denotes individual fixed effects, $\sigma_t$ denotes year fixed effects, and $\varepsilon_{it}$ is a residual term. $\beta$ as the coefficient of interest in this paper, if it is significantly negative, it indicates that co-ordination advancement of two-way FDI has a positive effect on resource misallocation, but vice versa worsens the level of resource misallocation.

## 3.2 Variable declaration

Explained variables: capital resource misallocation exponent(*abstauk*) and labor resource misallocation exponent (*abstaul*). Borrowing from the methodology proposed by Bai and Liu [8], the capital resource misallocation index (*abstauk*) and labor misallocation index (*abstaul*) can be computed.

First, the following model is derived by constructing the Cobb-Douglas production function and dividing both sides by the logarithm of the quantity of labor:

$$\ln\left(\frac{Y_{it}}{L_{it}}\right) = \ln A + \omega_{ki} \ln\left(\frac{K_{it}}{L_{it}}\right) \tag{3}$$

Here, $\frac{Y_{it}}{L_{it}}$ is the GDP per capita of the *i* city in year *t*. $\frac{K_{it}}{L_{it}}$ is the average human fixed capital stock in year *t* for city *i*. Use of the perpetual inventory method to calculate the annual fixed capital stock of municipalities ($K_{it}$), the specific formula is as follows:

$$K_{it} = \frac{I_{it}}{P_{it}} + (1 - \rho_t)K_{t-1} \tag{4}$$

$K_{it}$ is the fixed capital stock of the *i*-th city in year *t*. $\frac{I_{it}}{P_{it}}$ is the fixed capital stock of each city in year *t* without depreciation rate. $\rho$ is the depreciation rate, which is generally 9.6 per cent, in accordance with common domestic practice.

Based on Eq (3), introducing variable coefficient models $\ln\left(\frac{K_{it}}{L_{it}}\right)$ with interaction terms for each city in the model and regression utilizing the LSDV approach provides coefficients that are estimated to be distinct across individuals in each cross-section. That is $\omega_{ki}$, which will be brought into the following equation:

$$\gamma_{Ki} = \left(\frac{K_i}{K}\right) \Big/ \left(\frac{S_i \omega_{ki}}{\omega_k}\right), \gamma_{Li} = \left(\frac{L_i}{L}\right) \Big/ \left(\frac{S_i \omega_{Li}}{\omega_L}\right) \tag{5}$$

Where, $\gamma_{Ki}, \gamma_{Li}$ denote the coefficient of capital price distortion and the coefficient of labour price distortion, respectively; $\frac{K_i}{K}$ and $\frac{L_i}{L}$ denote the ratio of capital to total capital and labor to

total labor, respectively. $\frac{S_i \omega_{Ki}}{\omega_K}$ and $\frac{S_i \omega_{Li}}{\omega_L}$ denote the proportion of capital and the proportion of labor used by city $i$ when capital and labor are efficiently allocated, respectively. When $\gamma_{Ki}$ and $\gamma_{Ki}$ are greater than 1, which indicates that city $i$ is over-resourced for that year, and when it is less than 1 it indicates an under-allocation of resources. The capital and labor mismatch exponent are then calculated by the following equation:

$$tauk_{it} = \frac{1}{\gamma_{Ki}} - 1, \quad taul_{it} = \frac{1}{\gamma_{Li}} - 1 \tag{6}$$

$tauk_{it}$ and $taul_{it}$ denote the capital resource misallocation and labor resource misallocation indices, respectively. The computed indices for mismatched capital and labor may be positive or negative, indicating the economic significance of under- or over-allocation of these resources. The greater the absolute value, the more severe the degree of such under- or over-allocation. From this, the absolute values are used to obtain the new capital resource misallocation exponent (*abstauk*) and labor misallocation exponent (*abstaul*) [23]. The higher the value, the lower the efficiency of resource allocation and vice versa a more efficient allocation of resource.

Core explanatory variable: level of the co-ordination advancement of two-way FDI (*dioc*), Huang et al. [31], a metric has been devised to evaluate the coordination between urban IFDI and OFDI.

Outward foreign direct investment (OFDI) refers to the transfer of capital from the home country to the host country for production and sales by the mode of overseas operation and management. In this paper, it refers to China's foreign direct investment; Inward foreign direct investment (IFDI) refers to the process by which foreign enterprises or individuals inject capital, technology, management experience and other resources into the economic system of a specific country. In this paper, it refers to the investment made by foreign enterprises or individuals in China.

The specific calculations are as follows: First, IFDI data for each city was obtained from the *city's statistical yearbook for the calendar year*. Second, the OFDI value of each province in the calendar year is obtained through the *'China OFDI Bulletin'*, and then the OFDI data of each city in the calendar year is estimated based on the proportion of the GDP of each city to the GDP of the province. Third, the coupling degree of two-way FDI is calculated using the coupled system model of physics with the following equation:

$$dioc_{it} = \left[ (C_{it}(OI) * (ofdi_{it} + ifdi_{it}) / 2 \right]^{1/2} \tag{7}$$

$$C_{it}(IO) = IFDI_{it} * OFDI_{it} / (\mu IFDI_{it} + \theta OFDI_{it})^{\varphi} \tag{8}$$

$dioc_{it}$ is coordinated enhancement level for two-way FDI; $C_{it}(IO)$ is the coupling formula; $IFDI_{it}$ and $OFDI_{it}$ denote IFDI, OFDI flows for city $i$ in year $t$, respectively. The parameters μ, θ,φ are the share of IFDI, the share of OFDI and the adjustment factor. Referring to Huang et al. [31] practice, set them to 0.5, 0.5, and 2, respectively.

The following control variables are selected: level of regional economic enhancement (ln*pgdp*), the allocation of regional resources is influenced by the level of local economic development [32], as measured by the ln*pgdp*. In general, as the advancement of economics, the ability to allocate resources is greater, this paper takes the logarithm of GDP per capita to measure the level of regional economic enhancement. Level of regional public infrastructure (*pub*), the natural logarithm of the total public book collection in prefecture-level cities was used to measure the level of public infrastructure development [33]; Level of financial technology (*fint*) approach, the level of fintech enhancement is measured by constructing a keyword

**Table 1. Main variables descriptive statistics.**

| Variables | Obs. | Mean | St. D | Min. | Max |
|---|---|---|---|---|---|
| *abstauk* | 3341 | 0.278 | 0.301 | 0 | 4.017 |
| *abstaul* | 3341 | 0.458 | 0.387 | 0 | 3.212 |
| *dioc* | 3341 | 0.462 | 0.112 | 0.059 | 0.754 |
| ln*pgdp* | 3341 | 15.18 | 1.531 | 5.333 | 20.91 |
| *pub* | 3341 | 7.294 | 1.051 | 3.664 | 11.83 |
| *fint* | 3341 | 2.167 | 1.614 | 0 | 6.841 |
| ln*fin* | 3341 | 1.568 | 1.552 | 0.0270 | 20.68 |

library of fintech and then using the Baidu exponent to obtain the number of fintech search entries for each city, which is taken as the natural logarithm [34]. Financial expenditures (ln*fin*), the degree of government intervention in economic activity can have an impact on regional resource mismatch [35]. In this paper, the natural logarithm of expenditures within the general budget of local finances is chosen to measure government intervention.

There are three main data sources for this paper: first, data from the *China Outward Investment Bulletin*; second, *China Urban Statistical Yearbook*; and third, *China Regional Economic Statistical Yearbook*. China started releasing OFDI data in 2003, although it was not until 2006 that the information for the first few years, which had been more seriously incomplete, became relatively comprehensive. Meanwhile, the *Outward Investment Bulletin* is currently only updated to 2020, with only 2019 data available. Therefore, the study period for this paper was established as 2006–2019. In addition, considering the existence of mergers, splits, etc. in some prefecture-level cities, the final data in this paper covers only 285 cities. Table 1 shows main variables descriptive statistics.

## 4 Empirical analyses

### 4.1 Benchmark regression

This paper's data are panel data. Drawing on [36], Before conducting empirical research, we have to use the Hausman test to choose the choice between fixed and random effects. The result shows are chi2 (5) = 46.41, Prob > chi2 = 0.000, we need to use a fixed effect. Table 2 presents the estimation results of the baseline regression. The first two columns examine the co-ordination advancement of two-way FDI's influence on capital resource misallocation and labor resource misallocation, respectively, when fixing time and urban. The findings are the co-ordination advancement of two-way FDI festers capital resource misallocation at a significance level of 1%. Furthermore, the estimated coefficient for labor misallocation is significantly positive at the same level of significance. The co-ordination advancement of two-way FDI is conducive to correct capital resource misallocation and an adverse effect on labor misallocation. Considering that co-ordination advancement of two-way FDI may have a time lag e influence on capital allocation and labor resource allocation. Therefore, in the last two columns of Table 2, the influence of the lag in the co-ordination advancement of two-way FDI on the mis allocation between capital and labor is examined. In addition, the estimated result shows little change compared to the first two columns, indicating that the co-ordination advancement of two-way FDI has a relatively stable influence on capital resource misallocation and labor misallocation. H1 was tested.

Among the control variables, the level of regional economic enhancement exerts a significant deteriorating impact on labor misallocation while showing no impact on capital resource misallocation. The level of regional public infrastructure has a significant mitigating effect on

**Table 2. Benchmark regression.**

|  | *abstauk* | *abstaul* | *abstauk* | *abstaul* |
|---|---|---|---|---|
|  | **(1)** | **(2)** | **(3)** | **(4)** |
| *dioc* | -0.0252*** | 0.0757*** |  |  |
|  | (0.0076) | (0.0127) |  |  |
| *L.dioc* |  |  | -0.0256*** | 0.0698*** |
|  |  |  | (0.0075) | (0.0132) |
| ln*pgdp* | 0.0020 | 0.0380*** | -0.0011 | 0.0414*** |
|  | (0.0065) | (0.0102) | (0.0061) | (0.0110) |
| *pub* | -0.0319*** | -0.0307*** | -0.0345*** | -0.0275*** |
|  | (0.0064) | (0.0099) | (0.0066) | (0.0101) |
| *fint* | 0.0026 | -0.0164** | 0.0035 | -0.0114 |
|  | (0.0043) | (0.0074) | (0.0043) | (0.0079) |
| ln*fin* | 0.0111* | 0.0059 | 0.0034 | 0.0105* |
|  | (0.0058) | (0.0048) | (0.0047) | (0.0057) |
| _cons | 0.5830*** | -0.2174 | 0.6661*** | -0.2792 |
|  | (0.1076) | (0.1703) | (0.1046) | (0.1870) |
| Year FE | Y | Y | Y | Y |
| City FE | Y | Y | Y | Y |
| N | 3810 | 3810 | 3556 | 3556 |
| adj. $R^2$ | 0.763 | 0.695 | 0.785 | 0.700 |
| F | 7.8563 | 15.3110 | 8.7177 | 11.9801 |

Notes: Values in brackets indicate standard errors for city clustering

*, **, *** indicate significance at 10%, 5%, and 1%, respectively, as shown below.

the labor misallocation and capital resource misallocation. The level of financial technology significantly reduces labor misallocation in the current period, but has no significant effect on capital resource misallocation. The level of local finance deteriorates capital resource misallocation and labor misallocation, but it's not very significant.

## 4.2 Robustness tests

The capital deformation coefficient (*kcon*) and labor deformation coefficient (*lcon*) are calculated using Bai and Liu [8] methodology and capital deformation coefficients and labor deformation coefficients are used instead of capital resource misallocation and labor resource misallocation. The capital and labor deformation coefficients are calculated as:

$$kcon = \frac{MP_K}{r} - 1 = \beta K_i = \frac{p_i y_i}{r k_i} - 1 \tag{9}$$

$$lcon = \frac{MP_L}{w} - 1 = \beta L_i = \frac{p_i y_i}{w_i k_i} - 1 \tag{10}$$

$p_i y_i$ is the total value of production; $r$ is capital's price. Hsieh and Klenow [5] a 10% value was obtained; $W_i$ is the price of labor in city $i$. MP$_k$ and $MP_l$ are the marginal output of capital and labor, respectively.

Constructing the coefficient of capital distortion as well as the coefficient of labor distortion according to Eqs 9 and 10. Table 3 shows the calculation results of the impact of the co-ordination advancement of two-way FDI levels on the deformation coefficient of capital and labor.

**Table 3. Robustness test.**

| | kcon | kcon | lcon | lcon |
|---|---|---|---|---|
| | (1) | (2) | (3) | (4) |
| dioc | -0.0569*** | | 0.0336** | |
| | (0.0162) | | (0.0149) | |
| L.dioc | | -0.0440** | | 0.0359* |
| | | (0.0182) | | (0.0184) |
| lnpgdp | 0.0171 | 0.0159 | 0.0524*** | 0.0519*** |
| | (0.0108) | (0.0110) | (0.0070) | (0.0076) |
| pub | 0.0022 | 0.0028 | -0.0012 | -0.0036 |
| | (0.0136) | (0.0140) | (0.0129) | (0.0135) |
| fint | 0.0099 | 0.0069 | -0.0374*** | -0.0343*** |
| | (0.0095) | (0.0099) | (0.0114) | (0.0116) |
| lnfin | -0.0023 | 0.0003 | -0.0221*** | -0.0212*** |
| | (0.0067) | (0.0069) | (0.0062) | (0.0062) |
| _cons | 1.1296*** | 1.0919*** | -0.0408*** | 1.0245*** |
| | (0.1947) | (0.1988) | (0.0093) | (0.1471) |
| Year FE | Y | Y | Y | Y |
| City FE | Y | Y | Y | Y |
| N | 3810 | 3810 | 3570 | 3315 |
| adj. $R^2$ | 0.699 | 0.716 | 0.746 | 0.750 |
| F | 3.0930 | 1.5379 | 244.3748 | 236.3851 |

Notes: Values in brackets indicate standard errors for city clustering

*, **, *** indicate significance at 10%, 5%, and 1%, respectively, as shown below.

Column (2) show the results of estimating the coefficient of capital distortion in the current and lagged a period for the co-ordination enhancement of two-way FDI. It can be seen the negative coefficient impact on capital distortion is significant for both the current and lagged periods of the co-ordination advancement of two-way FDI levels. This indicates that it can significantly alleviate the degree of capital resource misallocation. The last two columns show the impact of the co-ordination advancement of two-way FDI on labor deformation coefficient during the current and lagging period. The results suggest that the research results indicate that the coordinated enhancement level of current and lagged two-way FDI has a promoting effect on the coefficient of labor distortion.

## 4.3 Endogeneity test

The co-ordination advancement of two-way FDI and resource misallocation are both important economic variables within the economic system, and their endogeneity is inevitable. That is the co-ordination advancement of two-way FDI impacts resource misallocation, and vice versa. Furthermore, there are also issues of missing variables and measurement errors in the model, which are the main sources of endogeneity. To overcome the interference of potential endogeneity on the estimation results, this paper uses two approaches to deal with it:

First, the instrumental variables approach. Bartik [37] developed these instrumental variables. Specifically, the method involves multiplying the initial value of each city's degree of the co-ordination advancement of two-way FDI by the growth rate of the national average degree of the co-ordination advancement of two-way FDI relative to the initial period, resulting in the Bartik instrumental variable (*bartikiv*). This instrumental variable demonstrates a strong

**Table 4. Estimation of instrumental variables.**

| | abstauk | abstauk | abstaul | abstaul |
|---|---|---|---|---|
| | (1) | (2) | (3) | (4) |
| dioc | -0.3496*** | | 0.1585*** | |
| | (0.0344) | | (0.0387) | |
| L.dioc | | -0.3528*** | | 0.1456*** |
| | | (0.0340) | | (0.0393) |
| lnpgdp | 0.0495*** | 0.0354*** | 0.0258** | 0.0330*** |
| | (0.0091) | (0.0085) | (0.0102) | (0.0098) |
| pub | -0.0130 | -0.0179* | -0.0356*** | -0.0313*** |
| | (0.0092) | (0.0092) | (0.0104) | (0.0107) |
| fint | 0.0161** | 0.0142** | -0.0198*** | -0.0139* |
| | (0.0064) | (0.0064) | (0.0072) | (0.0074) |
| lnfin | 0.0319*** | 0.0262*** | 0.0006 | 0.0052 |
| | (0.0044) | (0.0047) | (0.0050) | (0.0054) |
| Year FE | Y | Y | Y | Y |
| City FE | Y | Y | Y | Y |
| N | 3810 | 3556 | 3810 | 3556 |
| The first stage | | | | |
| bartikiv | 0.6665*** | | 0.6665*** | |
| | (0.0429) | | (0. 0429) | |
| L.bartikiv | | 0.7142*** | | 0.7142*** |
| | | (0.0446) | | (0.0446) |
| Underidentification test (Anderson canon. corr. LM statistic) | 226.569 | 239.181 | 226.569 | 239.181 |
| Weak identification test (Cragg-Donald Wald F statistic) | 240.694 | 256.454 | 240.694 | 256.454 |

Notes: The Table 4 displays regression coefficients with standard errors in parentheses. Statistical significance is denoted by

*, **, ***, which correspond to $p < 0.1$, $p < 0.05$, and $p < 0.01$, respectively. This notation is consistent throughout the text.

correlation with the co-ordination advancement of two-way FDI, but a relatively weak correlation with the level of resource misallocation (as measured by capital resource misallocation indices and labor misallocation indices). Table 4 presents the estimation results of the instrumental variables. Notably, there are no estimates indicating weak instrumental variables or overidentification in the first stage. Additionally, both the estimates of *bartikIV* and L.*bartikIV* are highly significant and positive. The findings from the second-stage estimation suggest that the co-ordination advancement of two-way FDI has a moderating effect on capital resource misallocation and exacerbates labor misallocation, consistent with the results obtained from the benchmark regression estimation.

Second, system GMM estimation is utilized. Systematic GMM estimation can incorporate lagged explanatory variable and instrumental variables effectively, enabling the selection of endogenous variables, making it a useful method for addressing potential endogeneity. Employing the system GMM approach, this paper utilizes the *xtdpdsys* command, considering *dioc* as an endogenous variable. The results of the estimation using the two-step method are presented in Table 5. The findings indicate that the coefficients of the previous timeframe for both capital resource misallocation and labor resource misallocation are significantly positive, suggesting a path dependence between capital resource misallocation and labor resource misallocation over time. The calculated coefficients reveal a notably negative correlation between capital resource misallocation and both the coordination advancement of two-way FDI and the lagged one-period coordination enhancement of two-way FDI. Additionally, the estimated

**Table 5. System GMM estimation.**

|  | abstauk | abstauk | abstaul | abstaul |
|---|---|---|---|---|
|  | (1) | (2) | (3) | (4) |
| L.abstauk | 1.1121*** | 1.0951*** |  |  |
|  | (0.0040) | (0.0057) |  |  |
| L.abstaul |  |  | 0.9636*** | 1.0157*** |
|  |  |  | (0.0013) | (0.0045) |
| dioc | -0.0168*** |  | 0.0358*** |  |
|  | (0.0022) |  | (0.0009) |  |
| L.dioc |  | -0.0081** |  | 0.0105** |
|  |  | (0.0034) |  | (0.0053) |
| trade | 0.0114*** | 0.0135*** | -0.0012 | 0.0057* |
|  | (0.0018) | (0.0026) | (0.0011) | (0.0029) |
| book | 0.0087*** | 0.0105*** | -0.0089*** | 0.0221*** |
|  | (0.0024) | (0.0027) | (0.0014) | (0.0053) |
| fint | -0.0035*** | 0.0030** | 0.0043*** | 0.0001 |
|  | (0.0009) | (0.0015) | (0.0013) | (0.0031) |
| scie | 0.0010 | -0.0001 | 0.0046*** | 0.0038 |
|  | (0.0011) | (0.0008) | (0.0005) | (0.0023) |
| _cons | -0.1690*** | -0.3007*** | -0.0408*** | -0.1880*** |
|  | (0.0259) | (0.0372) | (0.0093) | (0.0408) |
| AR (1) P | 0.000 | 0.000 | 0.000 | 0.0000 |
| AR (2) P | 0.2041 | 0.2446 | 0.2080 | 0.2226 |
| Sargan P | 0.1887 | 0.1491 | 0.1837 | 0.1791 |
| N | 3556 | 3556 | 3556 | 3556 |

coefficients on labor misallocation are significantly positive for both the co-ordination advancement of two-way FDI and the co-ordination advancement of two-way FDI with a one-period lag. This is consistent with the initial estimates, and H1 is confirmed.

## 4.4 Heterogeneity test

This section examines the heterogeneity existing in both the regional and urban contexts.

First, the heterogeneity of the level of co-ordination advancement in two-way FDI, Chinese cities are classified based on the level of coordination advancement in two-way FDI, and the disparities in this regard are estimated. The first and third columns in Table 6 indicate that when the level of the co-ordination advancement of two-way FDI exceeds the average, it significantly improves capital mismatch while significantly exacerbating labor mismatch. This finding aligns with the benchmark regression results. The second and fourth columns show that when the level of the co-ordination advancement of two-way FDI is lower than the average level, the impact on resource allocation is insignificant due to the low level of the co-ordination advancement of two-way FDI.

Second, regional heterogeneity. Dividing China into the East the Central and the West. Estimations of regional differences in the impact of the co-ordination advancement of two-way FDI on capital resource misallocation are presented in the first three columns of Table 7. The results show that in the East and West regions, the co-ordination advancement of two-way FDI has a significant positive impact on capital resource misallocation, only in the middle of the country the impact is not significant. This may be due to the co-ordination advancement of two-way FDI in the eastern and western regions being at the highest and the lowest level,

**Table 6. Two-way FDI coordinated enhancement level heterogeneity.**

| | abstauk | | abstaul | |
|---|---|---|---|---|
| | High-coordination | Low-coordination | High coordination | Low-coordination |
| | (1) | (2) | (3) | (4) |
| lndioc | -0.0817*** | 0.0023 | 0.1759*** | 0.0212 |
| | (0.0308) | (0.0078) | (0.0321) | (0.0187) |
| trade | 0.0029 | 0.0029 | 0.0161 | 0.0344*** |
| | (0.0172) | (0.0072) | (0.0203) | (0.0126) |
| book | -0.0316*** | -0.0150** | -0.0172 | -0.0133 |
| | (0.0114) | (0.0076) | (0.0173) | (0.0140) |
| fint | -0.0014 | -0.0055 | -0.0310** | 0.0142 |
| | (0.0086) | (0.0046) | (0.0140) | (0.0092) |
| scie | 0.0121 | -0.0057 | -0.0126** | 0.0395 |
| | (0.0086) | (0.0055) | (0.0060) | (0.0252) |
| _cons | 0.9446*** | 0.3208*** | -0.4395 | -0.1410 |
| | (0.2947) | (0.1185) | (0.4188) | (0.1973) |
| Year FE | Y | Y | Y | Y |
| City FE | Y | Y | Y | Y |
| N | 1882 | 1889 | 1882 | 1889 |
| adj. $R^2$ | 0.8245 | 0.6307 | 0.7282 | 0.7260 |
| F | 3.6290 | 1.2486 | 6.8381 | 4.2513 |

which has a significant improvement effect on capital resource misallocation; The final three columns display how the co-ordination advancement of two-way FDI impacts on the regional differences in labor capital resource misallocation, revealing a highly negative effects for the eastern and western regions, but a non-significant negative effect in the central area. This

**Table 7. Regional heterogeneity.**

| | abstauk | | | abstaul | | |
|---|---|---|---|---|---|---|
| | Easten | Middle | Westen | Eaten | Middle | Westen |
| | (1) | (2) | (3) | (4) | (5) | (6) |
| dioc | -0.0575** | 0.0124 | -0.0424*** | 0.1330*** | -0.0064 | 0.0805*** |
| | (0.0227) | (0.0095) | (0.0101) | (0.0232) | (0.0289) | (0.0158) |
| lnpgdp | -0.0089 | 0.0142 | -0.0060 | 0.0437** | 0.0323** | 0.0600*** |
| | (0.0164) | (0.0114) | (0.0092) | (0.0192) | (0.0137) | (0.0193) |
| pub | -0.0939*** | -0.0053 | -0.0005 | 0.0018 | -0.0657*** | -0.0174 |
| | (0.0170) | (0.0078) | (0.0076) | (0.0153) | (0.0204) | (0.0159) |
| fint | 0.0198** | -0.0181*** | -0.0064 | -0.0242* | 0.0067 | -0.0167 |
| | (0.0093) | (0.0063) | (0.0058) | (0.0128) | (0.0144) | (0.0116) |
| lnfin | 0.0329*** | -0.0544*** | -0.0042 | -0.0058 | 0.0703*** | 0.0022 |
| | (0.0120) | (0.0082) | (0.0036) | (0.0062) | (0.0221) | (0.0062) |
| _cons | 1.3752*** | 0.1241 | 0.5632*** | -0.8041** | 0.3856* | -0.6956** |
| | (0.2657) | (0.1779) | (0.1494) | (0.3469) | (0.2297) | (0.3209) |
| Year FE | Y | Y | Y | Y | Y | Y |
| City FE | Y | Y | Y | Y | Y | Y |
| N | 1470 | 975 | 1365 | 1470 | 975 | 1365 |
| adj. $R^2$ | 0.813 | 0.577 | 0.545 | 0.703 | 0.699 | 0.693 |
| F | 10.4669 | 11.3415 | 4.5425 | 7.4774 | 4.4816 | 7.8711 |

**Table 8. Urban hierarchical heterogeneity.**

| | abstauk | | | abstaul | | |
|---|---|---|---|---|---|---|
| | First-tier and new first-tier cities | Second and third tier cities | Fourth-tier cities | First-tier and new first-tier cities | Second and third tier cities | Fourth-tier cities |
| | (1) | (2) | (3) | (4) | (5) | (6) |
| dioc | -0.6055*** | -0.0439*** | -0.0383*** | -0.0613* | 0.0980*** | 0.0653*** |
| | (0.0899) | (0.0117) | (0.0139) | (0.0366) | (0.0179) | (0.0183) |
| ln*pgdp* | 0.1334 | -0.0061 | 0.0026 | 0.0059 | 0.0567*** | 0.0546*** |
| | (0.1443) | (0.0108) | (0.0125) | (0.0352) | (0.0171) | (0.0168) |
| pub | -0.1205** | -0.0239** | -0.0180 | 0.0165 | 0.0062 | -0.0093 |
| | (0.0473) | (0.0102) | (0.0125) | (0.0412) | (0.0143) | (0.0148) |
| fint | -0.1380*** | 0.0084 | 0.0011 | 0.0258 | -0.0574*** | -0.0335*** |
| | (0.0442) | (0.0065) | (0.0083) | (0.0212) | (0.0125) | (0.0119) |
| ln*fin* | 0.0067 | -0.0029 | -0.0094** | 0.0023 | -0.0132* | 0.0132 |
| | (0.0192) | (0.0038) | (0.0040) | (0.0057) | (0.0077) | (0.0105) |
| _cons | 3.6134 | 0.7241*** | 0.5201*** | 0.5037 | -0.8113*** | -0.5430* |
| ln*pgdp* | (2.6928) | (0.1807) | (0.1990) | (0.7574) | (0.2942) | (0.2777) |
| Year FE | Y | Y | Y | Y | Y | Y |
| City FE | Y | Y | Y | Y | Y | Y |
| N | 285 | 1470 | 1020 | 285 | 1470 | 1020 |
| adj. $R^2$ | 0.882 | 0.630 | 0.580 | 0.852 | 0.680 | 0.766 |
| F | 14.9564 | 5.4104 | 4.1220 | 0.9667 | 11.4767 | 8.3237 |

could be attributed to an abundance of labor in the East and a shortage of labor in the West, with reciprocal the co-ordination advancement of two-way FDI further exacerbating the labor misallocation.

Third, urban heterogeneity. China is divided into first-line, new first-line, second-line, third-line, and fourth-tier cities based on the *New Classification List of Chinese Cities*. The empirical results are shown in Table 8. It is evident that the co-ordination advancement of two-way FDI positive impacts on capital resource misallocation in all types of cities, including first-tier and new first-tier, second- and third-tier, and fourth-tier cities. This indicates that the co-ordination advancement of two-way FDI at any city level has a mitigating effect on capital resource misallocation, implying that this effect diminishes as the city gets smaller. The impact of the co-ordination advancement of two-way FDI on labor misallocation is improved in line 1 and new line 1 cities, but is exacerbated in line 2, line 3 and line 4 cities, and the larger city, the greater exacerbation.

## 5 Mechanism analysis

The mediating effect was used to validate the mechanism in this paper (Cui et al., 2019) [38]. Firstly, this article uses the natural logarithm of the balance of deposits and loans in the financial sector to calculate the enhancement level of the financial industry (ln*finance*) and measures labor costs by means of the natural logarithm of urban per capita wage levels (ln*salary*). Secondly, a two-step approach is used to examine the extent of the context of the co-ordination advancement of two-way FDI on the level of enhancement of the financial sector and the cost of labor, and then the impact of extent of enhancement of the financial sector and the cost of labor on the misallocation both capital and labor. Finally, the study employs a three-step approach to examine the effect of the co-ordination advancement of two-way FDI on the level

**Table 9. Capital resource misallocation mechanisms.**

| | lnfinance | abstauk | abstauk |
|---|---|---|---|
| | **(1)** | **(2)** | **(3)** |
| dioc | 0.0918*** | | -0.0156** |
| | (0.0109) | | (0.0076) |
| lnfinance | | -0.0831*** | -0.1051*** |
| | | (0.0253) | (0.0291) |
| lnpgdp | 0.0659*** | 0.0069 | 0.0090 |
| | (0.0117) | (0.0067) | (0.0072) |
| pub | -0.0046 | -0.0300*** | -0.0324*** |
| | (0.0076) | (0.0063) | (0.0064) |
| fint | -0.0229*** | 0.0077* | 0.0002 |
| | (0.0052) | (0.0042) | (0.0042) |
| lnfin | -0.0025 | 0.0092* | 0.0109* |
| | (0.0041) | (0.0049) | (0.0056) |
| _cons | 16.0386*** | 1.8071*** | 2.2694*** |
| | (0.1833) | (0.4224) | (0.4816) |
| Year FE | Y | Y | Y |
| City FE | Y | Y | Y |
| N | 3810 | 3810 | 3810 |
| adj. $R^2$ | 0.981 | 0.754 | 0.766 |
| F | 29.1597 | 6.7991 | 7.1770 |

of financial sector enhancement and labor costs based on the benchmark regression, and then to examine the impact on capital and labor misallocation by simultaneously examining the co-ordination advancement of two-way FDI added to the model along with the level of financial sector enhancement and labor costs.

Table 9 shows the estimated results of the capital mismatch mechanism, the first column shows the effect of the co-ordination advancement of two-way FDI on the enhancement level of the financial sector and the consequence show that the co-ordination advancement of two-way FDI has made significant contributions to the improve the level of the financial sector. The effect of the level of financial sector enhancement on capital resource misallocation is shown in column (2). The level of enhancement of the financial sector significantly improves capital resource misallocation. Two-step estimation shows that the level of the co-ordination advancement of two-way FDI corrects capital resource misallocation through the level of enhancement of the financial sector. Column (3) is results of the third step, the coefficient in column (3) is -0.0156, which is a decrease in absolute value compared to the coefficient of -0.0252 in column (2) of the baseline regression Table 2, and the coefficient on the level of enhancement of the financial sector is negative at this point. Multiplying by 0.0918 in column (1) and then dividing by -0.0252 results in a mediation effect size of 38.3%, indicating a partial mediation effect. Hypothesis 1 is validated by the analyses in Table 9.

Similarly, Table 10 presents estimated consequences of the labor misallocation mechanism. The first column illustrates how the co-ordination advancement of two-way FDI influences labor costs and the findings indicate a significant increase in labor costs due to such enhance-ment. The second column illustrates the what labor costs influence on labor misallocation, and the consequences indicate that the increase in labor costs has a serious negative impact on labor misallocation The outcomes of the two-step estimation illustrate that the co-ordination advancement of two-way FDI exacerbates labor misallocation by escalating labor costs.

**Table 10. Analysis of labor misallocation mechanisms.**

| | ln*salary* | *abstaul* | *abstaul* |
|---|---|---|---|
| | **(1)** | **(2)** | **(3)** |
| *dioc* | 0.0294*** | | 0.0644*** |
| | (0.0050) | | (0.0147) |
| ln*salary* | | 0.2251*** | 0.2078*** |
| | | (0.0515) | (0.0531) |
| ln*pgdp* | 0.0249*** | 0.0440*** | 0.0328*** |
| | (0.0062) | (0.0118) | (0.0119) |
| *pub* | -0.0081* | -0.0302*** | -0.0419*** |
| | (0.0043) | (0.0110) | (0.0108) |
| *fint* | -0.0048* | -0.0050 | -0.0154** |
| | (0.0027) | (0.0069) | (0.0072) |
| ln*fin* | 0.0007 | 0.0031 | -0.0007 |
| | (0.0016) | (0.0045) | (0.0044) |
| _cons | 0.9264*** | -0.2821 | -0.2818 |
| | (0.0969) | (0.1998) | (0.1950) |
| Year FE | Y | Y | Y |
| City FE | Y | Y | Y |
| N | 3810 | 3810 | 3810 |
| adj. $R^2$ | 0.974 | 0.734 | 0.737 |
| F | 13.6488 | 8.7055 | 13.3928 |

Column (3) shows that no matter the co-ordination advancement of two-way FDI or labor costs worsen labor misallocation. The mediation effect is calculated to be 8.75%, indicating that there is a mediation effect. During the demographic dividend period, China benefited significantly from the co-ordination advancement of two-way FDI thanks to its low-cost labor. However, with increased production technology, the demand for low and medium-skilled labor has decreased, resulting in a reduction in the number of labor positions. This has worsened labor mismatch issues. Hypothesis 2 is validated by the analyses in Table 10.

## 6 Conclusions and recommendations

With the continuous promotion of the dual-cycle strategy and the guidance of the 'bringing in' and 'going out' policy, strategic support has been provided for achieving high-quality enhancement in the new era. This paper integrates the advancement of two-way FDI coordination and resource misallocation into the same analytical framework. It calculates the degree of coupling coordination and indicators representing the extent of resource misallocation caused by two-way FDI in prefectural-level cities from 2006 to 2019, and adopts methods such as bidirectional fixed effects, instrumental variables, and system GMM to examine the impacts and mechanism of the co-ordination advancement of two-way FDI on capital and labor misallocation. What the study found: Firstly, the enhancement of capital resource misallocation due to the co-ordination advancement of two-way FDI in China, coupled with the exacerbation of labor resource misallocation, persists after rigorous testing; Secondly, the co-ordination advancement of two-way FDI alleviates capital resource misallocation but exacerbates labor misallocation in the Eastern and Western regions. The co-ordination advancement of two-way FDI improves capital resource misallocation across all types of cities, while exacerbating labor misallocation primarily in 2-tier, 3-tier, and 4-tier cities. The co-ordination advancement of two-way FDI improves capital resource misallocation across all types of cities, However, it

primarily exacerbates labor resource misallocation in second-tier, third-tier, and fourth-tier cities."

Based on these conclusions, the paper makes the following proposals: Firstly, at the national level, there should be continued promotion of the "bring in" and "go out" strategy, along with the co-ordination advancement of two-way FDI. This can help improve capital resource misallocation through the co-ordination advancement of two-way FDI. Secondly, at the local level, it is crucial to address the deteriorating labor misallocation resulting from the co-ordination advancement of two-way FDI. Enterprises must be actively encouraged to invest and create more job opportunities, particularly for surplus rural labor, to alleviate the labor misallocation issue. Thirdly, regional wage levels should be appropriately managed to alleviate the problem of regional labor misallocation. Over the past decade, wage levels in various regions have risen rapidly due to factors such as the disappearance of the demographic dividend and the increased intensity and technical difficulty of specific jobs. Both of these have led to an increase in wage levels. Faced with continuously rising wage levels, reducing domestic investment and turning to OFDI could harm the enhancement of the regional economy. Fourthly, the enhancement of the regional financial sector must be enhanced to provide enterprises with additional financing channels to address the capital resource misallocation issue across the region. The hindrance of enterprise financing has consistently been a crucial factor impeding enterprises from investing. Improving and developing the financial sector in districts can alleviate financing constraints to some extent, providing investment opportunities for enterprises and encouraging the co-ordination advancement of two-way FDI foreign direct investment. Lastly, at the macro level, outbound direct investment and foreign direct investment should be controlled to curb illegal capital outflow and provide a better environment for optimizing the allocation of domestic resources [39].

## Supporting information

**S1 Data.**
(ZIP)

## Acknowledgments

We thank those anonymous reviewers and the editor whose comments/suggestions helped improve and clarify this manuscript.

## Author Contributions

**Conceptualization:** Suqing Shao.

**Data curation:** Shuo Zhou.

**Formal analysis:** Shuo Zhou.

**Investigation:** Shuo Zhou, Suqing Shao.

**Methodology:** Shuo Zhou, Suqing Shao.

**Project administration:** Shuo Zhou.

**Resources:** Shuo Zhou.

**Software:** Shuo Zhou.

**Supervision:** Suqing Shao.

**Visualization:** Shuo Zhou.

**Writing – original draft:** Shuo Zhou.

**Writing – review & editing:** Suqing Shao.

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
