## [Decision Letter · Decision Letter 0]

25 Jan 2024

PONE-D-23-36401Can co-ordination advancement of two-way FDI Improve Resource Misallocation? Evidence from 285 Cities in ChinaPLOS ONE

Dear Dr. Shao,

Thank you for submitting your manuscript to PLOS ONE. After careful consideration, we feel that it has merit but does not fully meet PLOS ONE’s publication criteria as it currently stands. Therefore, we invite you to submit a revised version of the manuscript that addresses the points raised during the review process.

 **The two reviewers have provided professional comments, most of which focus on the empirical part. In this round, the authors should better enhance the empirical part to respond to the comments of the two reviewers.**

We look forward to receiving your revised manuscript.

Kind regards,

Wei Liu

Academic Editor

PLOS ONE

Journal Requirements:

Key Projects of Anhui Provincial University Research (2023AH050247)

Reviewers' comments:

Reviewer's Responses to Questions

**Comments to the Author**

1. Is the manuscript technically sound, and do the data support the conclusions?

Reviewer #1: Partly

Reviewer #2: Partly

2. Has the statistical analysis been performed appropriately and rigorously? 

Reviewer #1: Yes

Reviewer #2: Yes

3. Have the authors made all data underlying the findings in their manuscript fully available?

Reviewer #1: Yes

Reviewer #2: Yes

4. Is the manuscript presented in an intelligible fashion and written in standard English?

Reviewer #1: Yes

Reviewer #2: No

5. Review Comments to the Author

Reviewer #1: 1.The analysis of labor force mismatch mechanism does not match the meaning of labor force mismatch

2.Lack of endogeneity testing

3.Heterogeneity can be achieved at different levels of coordinated promotion of bidirectional FDI

Reviewer #2: Can co-ordination advancement of two-way FDI Improve Resource Misallocation? Evidence from 285 Cities in China

Referee’s comments

My comments are as follows;

• The abstract needs to be rewritten to systematically provide a proper summary of the study touching on the objective, the method, the timeframe, the findings, the conclusion and recommendations.

• The manuscript requires language editing.

• Abstauk and abstaul are not explanatory variables in equations 1 and 2.

• The econometric model (including all the control variables) that was eventually estimated is not provided.

• IFDI is not clearly defined, does this refer to FDI inflows into China?

• The study does not properly explain the theoretical link between fdi flows and resource misallocation. This should be the corner stone of this study.

• Which cities were covered in the study?

• The literature should be improved. Authors should incorporate the following papers along with other recent studies on the issue.

Illicit financial outflows, informal sector size and domestic resource mobilization in selected African countries

External financing for inclusive growth in lower-middle income West African countries: foreign direct investment versus official development assistance

• The estimation techniques used should be properly discussed with relevant equations stated.

6. PLOS authors have the option to publish the peer review history of their article (what does this mean?). If published, this will include your full peer review and any attached files.

Reviewer #1: No

Reviewer #2: No

---

## [Author Response · Author response to Decision Letter 0]

11 Mar 2024

Response to Reviewers

First of all, we would like to thank reviewers for your insightful, constructive, and helpful comments on our manuscript entitled “Can co-ordination advancement of two-way FDI Improve Resource Misallocation? Evidence from 285 Cities in China” We have carefully considered and addressed all the comments and made necessary revisions in the revised manuscript. We provide a point-by-point response to the reviewers’ comments below.

The points raised by the reviewers are written in bold font, whereas our responses are shown in normal font, and the quotation of the revised manuscript is shown in red font. 

Reviewer #1: 

 The analysis of labor force mismatch mechanism does not match the meaning of labor force mismatch.

Response:

Thank you very much for your question. Firstly, we have carefully considered your question. Do you mean that there are issues with the labor force mismatch mechanism in our theoretical hypothesis section and the subsequent labor force mismatch. Based on our understanding, we have revised the research hypothesis section. In revised manuscript, we propose two new research hypotheses to make the logic clearer: The first hypothesis focuses on the mechanism of capital mismatch, and the second is the mechanism of labor mismatch. The results of using the labor distortion coefficient in robustness testing are inconsistent. We have rerun the code and made modifications to the text and table.

…

2 Theoretical analysis and research hypotheses

Misallocation refers to a deviation from the ideal state of resource configuration. Unlike efficient resource allocation, 'resource misallocation' denotes the presence of barriers to the free allocation of different resource elements between regions and industries, hindering the achievement of optimal allocation. Cheng [24] classified resource misallocation into "internal mismatch" and "external mismatch". Bai and Liu [8] argue that varying degrees of capital misallocation and labor resource misallocation exist in different regions of China, with significant regional disparities. The following theoretical perspective analyses the effect of co-ordination advancement of two-way FDI on resource misallocation.

In recent years, foreign direct investment (FDI) and outward foreign direct investment (OFDI) have experienced rapid growth as two major strategic pathways for China to acquire funds, technology, and market resources. The trend of co-ordination advancement of two-way FDI is increasingly prominent, serving as a significant driving force in guiding the rational flow of resources, improving resource misallocation, and enhancing resource allocation efficiency. Specifically, the co-ordination advancement of two-way FDI can improve capital, with the main pathway being the level of financial development. On one hand, during the early stages of development in developing countries, IFDI can alleviate the host country's capital needs and bring advanced technology and management expertise [25]. Through vertical and horizontal effects, it helps address financing constraints for enterprises in the host country, thereby reducing financing costs, enhancing capital liquidity, fostering a more open, inclusive, and efficient financial system, consequently improving capital misallocation, and enhancing capital allocation efficiency [26]. On the other hand, capital inflows also provide a solid foundation for outward investments [27]. The enhancement of the host country's financial strength and the improvement of its financial development level create necessary conditions for enterprises to engage in cross-border capital allocation and operations. Through outward foreign direct investment (OFDI) reverse technology spillover effects, financial institutions in the host country have established close connections with those in developed countries. Through cooperation, exchange, and competition with financial institutions in these developed countries, the management level and operational efficiency of financial institutions in the host country have been significantly enhanced. Therefore, the interaction between foreign direct investment (FDI) and outward foreign direct investment (OFDI) continuously promotes and guides the improvement of the host country's financial development level. The improvement of the financial development level can increase the liquidity of resources, reduce transaction costs of financial investment, thereby increasing investment, effectively alleviating capital misallocation, and enhancing capital allocation efficiency [28]. 

H1: Coordinated development in two-way FDI can enhance capital, primarily through advancements in the level of financial development.

Coordinated development in two-way FDI exacerbates the mismatch of labor resources by increasing labor costs. On one hand, IFDI increases the demand for labor in the host country, leading to changes in the geographical distribution of labor and exacerbating the issue of labor mismatch. At the same time, foreign enterprises will attract sustained labor inflows by raising labor prices (Sun and Wang, 2022) [29]. This has led to a phenomenon of 'labor shortages' among local enterprises, exacerbating labor scarcity. With the continuous influx of IFDI, the host country may face challenges of labor supply shortages, further exacerbating resource misallocation. On the other hand, with the continuous outflow of OFDI, it may lead to the emigration of labor from some developing countries. This emigration of labor mainly manifests in two aspects. Firstly, some enterprises will export a portion of their labor force through outward foreign direct investment (Lai et al., 2020) [30]. Secondly, outward foreign direct investment companies will dispatch management and technical personnel to overseas subsidiaries, inevitably creating some low-skilled job positions. This, in turn, further exacerbates the host country's demand for labor. Due to the inability of labor supply to meet rapidly growing demand in the short term, the rise in labor costs is actually detrimental to addressing the issue of labor mismatch. Therefore, two-way FDI exacerbates the degree of labor mismatch.

H2: Coordinated development of two-way FDI exacerbates labor mismatch, primarily through increasing labor costs.

…

4.3 Robustness tests

The capital deformation coefficient (kcon) and labor deformation coefficient (lcon) are calculated using Bai and Liu [8] methodology and capital deformation coefficients and labor deformation coefficients are used instead of capital resource misallocation and labor resource misallocation. The capital and labor deformation coefficients are calculated as:

kcon=〖MP〗_K/r-1=〖βK〗_i=(p_i y_i)/〖rk〗_i -1 (9)

lcon=〖MP〗_L/w-1=〖βL〗_i=(p_i y_i)/〖w_i k〗_i -1 (10)

piyi is the total value of production; r is capital’s price. Hsieh and Klenow [5] a 10% value was obtained; Wi is the price of labor in city i. MPk and MPl are the marginal output of capital and labor, respectively.

Constructing the coefficient of capital distortion as well as the coefficient of labor distortion according to equations 9 and 10. Table 3 shows the calculation results of the impact of co-ordination advancement of two-way FDI on the deformation coefficient of capital and labor. Column (2) show the results of estimating the coefficient of capital distortion in the current and lagged a period for the co-ordination development of two-way FDI. It can be seen the negative coefficient impact on capital distortion is significant for both the current and lagged periods of co-ordination advancement of two-way FDI levels. This indicates that it can significantly alleviate the degree of capital misallocation. The last two columns show the impact of co-ordination advancement of two-way FDI on labor deformation coefficient during the current and lagging period. The results suggest that The research results indicate that the coordinated development level of current and lagged two-way FDI has a promoting effect on the coefficient of labor distortion.

Table 3. Robustness test

 kcon kcon lcon lcon

 (1) (2) (3) (4)

dioc -0.0569*** 0.0336** 

 (0.0162) (0.0149) 

L.dioc -0.0440** 0.0359*

 (0.0182) (0.0184)

lnpgdp 0.0171 0.0159 0.0524*** 0.0519***

 (0.0108) (0.0110) (0.0070) (0.0076)

pub 0.0022 0.0028 -0.0012 -0.0036

 (0.0136) (0.0140) (0.0129) (0.0135)

fint 0.0099 0.0069 -0.0374*** -0.0343***

 (0.0095) (0.0099) (0.0114) (0.0116)

lnfin -0.0023 0.0003 -0.0221*** -0.0212***

 (0.0067) (0.0069) (0.0062) (0.0062)

_cons 1.1296*** 1.0919*** -0.0408*** 1.0245***

 (0.1947) (0.1988) (0.0093) (0.1471)

Year FE Y Y Y Y

City FE Y Y Y Y

N 3810 3810 3570 3315

adj. R2 0.699 0.716 0.746 0.750

F 3.0930 1.5379 244.3748 236.3851

Notes: Values in brackets indicate standard errors for city clustering; *, **, *** indicate significance at 10%, 5%, and 1%, respectively, as shown below.

…

 Lack of endogeneity testing.

Response:

Thank you very much for your question. The original manuscript lacks essential endogeneity testing analysis, which will be supplemented in the revised version. The revised manuscript employs two methods to examine endogeneity. The first method involves employing instrumental variable techniques, where instrumental variables are generated to exhibit a strong correlation with the coordinated development of co-ordination advancement of two-way FDI but a weak correlation with levels of resource misallocation (capital misallocation index and labor misallocation index). This approach effectively addresses endogeneity concerns. The second approach involves employing the System GMM method to test for endogeneity issues, utilizing the differences of lagged variables to control for serial correlation. By employing these two methods, endogeneity interference is addressed. The revised part is in “4.4 Endogeneity test”.

…

4.4 Endogeneity test

The co-ordination advancement of two-way FDI and resource misallocation are both important economic variables within the economic system, and their endogeneity is inevitable. That is co-ordination advancement of two-way FDI impacts resource misallocation, and vice versa. Furthermore, there are also issues of missing variables and measurement errors in the model, which are the main sources of endogeneity. To overcome the interference of potential endogeneity on the estimation results, this paper uses two approaches to deal with it: 

First, the instrumental variables approach. Bartik [37] developed these instrumental variables. Specifically, the method involves multiplying the initial value of each city's degree of co-ordination advancement of two-way FDI by the growth rate of the national average degree of co-ordination advancement of two-way FDI relative to the initial period, resulting in the Bartik instrumental variable (bartikiv). This instrumental variable demonstrates a strong correlation with the co-ordination development of two-way FDI, but a relatively weak correlation with the level of resource misallocation (as measured by capital misallocation indices and labor misallocation indices). Table 4 presents the estimation results of the instrumental variables. Notably, there are no estimates indicating weak instrumental variables or overidentification in the first stage. Additionally, both the estimates of bartikIV and L.bartikIV are highly significant and positive. The findings from the second-stage estimation suggest that the co-ordination advancement of two-way FDI has a moderating effect on capital misallocation and exacerbates labor misallocation, consistent with the results obtained from the benchmark regression estimation.

Table 4. Estimation of instrumental variables

 abstauk abstauk abstaul abstaul

 (1) (2) (3) (4)

dioc -0.3496*** 0.1585*** 

 (0.0344) (0.0387) 

L.dioc -0.3528*** 0.1456***

 (0.0340) (0.0393)

lnpgdp 0.0495*** 0.0354*** 0.0258** 0.0330***

 (0.0091) (0.0085) (0.0102) (0.0098)

pub -0.0130 -0.0179* -0.0356*** -0.0313***

 (0.0092) (0.0092) (0.0104) (0.0107)

fint 0.0161** 0.0142** -0.0198*** -0.0139*

 (0.0064) (0.0064) (0.0072) (0.0074)

lnfin 0.0319*** 0.0262*** 0.0006 0.0052

 (0.0044) (0.0047) (0.0050) (0.0054)

Year FE Y Y Y Y

City FE Y Y Y Y

N 3810 3556 3810 3556

The first stage 

bartikiv 0.6665*** 0.6665*** 

 (0.0429) (0. 0429) 

L.bartikiv 0.7142*** 0.7142***

 (0.0446) (0.0446)

Underidentification test (Anderson canon. corr. LM statistic) 226.569 239.181 226.569 239.181

Weak identification test (Cragg-Donald Wald F statistic) 240.694 256.454 240.694 256.454

Notes: The table displays regression coefficients with standard errors in parentheses. Statistical significance is denoted by *, **, ***, which correspond to p < 0.1, p < 0.05, and p < 0.01, respectively. This notation is consistent throughout the text.

Second, system GMM estimation is utilized. Systematic GMM estimation can incorporate lagged explanatory variable and instrumental variables effectively, enabling the selection of endogenous variables, making it a useful method for addressing potential endogeneity. Employing the system GMM approach, this paper utilizes the xtdpdsys command, considering dioc as an endogenous variable. The results of the estimation using the two-step method are presented in Table 5. The findings indicate that the coefficients of the previous timeframe for both capital misallocation and labor resource misallocation are significantly positive, suggesting a path dependence between capital misallocation and labor resource misallocation over time. The calculated coefficients reveal a notably negative correlation between capital misallocation and both the coordination advancement of two-way FDI and the lagged one-period coordination development of two-way FDI. Additionally, the estimated coefficients on labor misallocation are significantly positive for both the coordination advancement of two-way FDI and the coordination advancement of two-way FDI with a one-period lag. This is consistent with the initial estimates, and H1 is confirmed.

Table 5. System GMM estimation

 abstauk abstauk abstaul abstaul

 (1) (2) (3) (4)

L.abstauk 1.1121*** 1.0951*** 

 (0.0040) (0.0057) 

L.abstaul 0.9636*** 1.0157***

 (0.0013) (0.0045)

dioc -0.0168*** 0.0358*** 

 (0.0022) (0.0009) 

L.dioc -0.0081** 0.0105**

 (0.0034) (0.0053)

trade 0.0114*** 0.0135*** -0.0012 0.0057*

 (0.0018) (0.0026) (0.0011) (0.0029)

book 0.0087*** 0.0105*** -0.0089*** 0.0221***

 (0.0024) (0.0027) (0.0014) (0.0053)

fint -0.0035*** 0.0030** 0.0043*** 0.0001

 (0.0009) (0.0015) (0.0013) (0.0031)

scie 0.0010 -0.0001 0.0046*** 0.0038

 (0.0011) (0.0008) (0.0005) (0.0023)

_cons -0.1690*** -0.3007*** -0.0408*** -0.1880***

 (0.0259) (0.0372) (0.0093) (0.0408)

AR (1) P 0.000 0.000 0.000 0.0000

AR (2) P 0.2041 0.2446 0.2080 0.2226

Sargan P 0.1887 0.1491 0.1837 0.1791

N 3556 3556 3556 3556

…

 Heterogeneity can be achieved at different levels of coordinated promotion of bidirectional FDI.

Response:

Thank you very much for your question. In the original manuscript, the examination of heterogeneity was conducted based on urban geographic locations and urban development levels, overlooking the impact of varying degrees of co-ordination advancement of two-way FDI on resource allocation.

In response to your feedback, we grouped cities based on different levels of co-ordination advancement of two-way FDI. Cities were divided into two groups: the first comprising cities with co-ordination advancement of two-way FDI levels above the average, and the second comprising cities with levels below the average. Regression analyses were then conducted within each group, and the obtained results remained significant. The revised part is in “4.5 Heterogeneity test”.

…

First, the heterogeneity of the level of co-ordination advancement in two-way FDI, Chinese cities are classified based on the level of coordination advancement in two-way FDI, and the disparities in this regard are estimated. The first and third columns indicate that when the level of coordination advancement in two-way FDI exceeds the average, it significantly improves capital mismatch while significantly exacerbating labor mismatch. This finding aligns with the benchmark regression results. Th

---

## [Decision Letter · Decision Letter 1]

21 May 2024

Can co-ordination advancement of two-way FDI Improve Resource Misallocation? Evidence from 285 Cities in China

PONE-D-23-36401R1

Dear Dr. Shao,

We’re pleased to inform you that your manuscript has been judged scientifically suitable for publication and will be formally accepted for publication once it meets all outstanding technical requirements.

Kind regards,

Wei Liu

Academic Editor

PLOS ONE

Additional Editor Comments (optional):

After the reviewers' comments, I think this paper can be accepted.

Reviewers' comments:

Reviewer's Responses to Questions

**Comments to the Author**

1. If the authors have adequately addressed your comments raised in a previous round of review and you feel that this manuscript is now acceptable for publication, you may indicate that here to bypass the “Comments to the Author” section, enter your conflict of interest statement in the “Confidential to Editor” section, and submit your "Accept" recommendation.

Reviewer #2: All comments have been addressed

2. Is the manuscript technically sound, and do the data support the conclusions?

Reviewer #2: Yes

3. Has the statistical analysis been performed appropriately and rigorously? 

Reviewer #2: Yes

4. Have the authors made all data underlying the findings in their manuscript fully available?

Reviewer #2: Yes

5. Is the manuscript presented in an intelligible fashion and written in standard English?

Reviewer #2: Yes

6. Review Comments to the Author

Reviewer #2: The authors have made the needed corrections. I am satisfied with the quality of the manuscript in its current form.

7. PLOS authors have the option to publish the peer review history of their article (what does this mean?). If published, this will include your full peer review and any attached files.

Reviewer #2: No

---

## [Editor Report · Acceptance letter]

31 May 2024

PONE-D-23-36401R1 

PLOS ONE

Dear Dr. Shao, 

I'm pleased to inform you that your manuscript has been deemed suitable for publication in PLOS ONE. Congratulations! Your manuscript is now being handed over to our production team.

Kind regards, 

on behalf of

Prof. Wei Liu 

Academic Editor

PLOS ONE